# Research and Diagnostic Algorithmic Rules (RADAR) and RADAR Plots for the First Episode of Major Depressive Disorder: Effects of Childhood and Recent Adverse Experiences on Suicidal Behaviors, Neurocognition and Phenome Features

**DOI:** 10.3390/brainsci13050714

**Published:** 2023-04-24

**Authors:** Michael Maes, Abbas F. Almulla

**Affiliations:** 1Department of Psychiatry, Faculty of Medicine, Chulalongkorn University, Bangkok 10330, Thailand; 2Cognitive Fitness and Technology Research Unit, Faculty of Medicine, Chulalongkorn University, Bangkok 10330, Thailand; 3Department of Psychiatry, Medical University of Plovdiv, 4002 Plovdiv, Bulgaria; 4Research Institute, Medical University Plovdiv, 4002 Plovdiv, Bulgaria; 5Kyung Hee University, 26 Kyungheedae-ro, Dongdaemun-gu, Seoul 02447, Republic of Korea; 6Medical Laboratory Technology Department, College of Medical Technology, The Islamic University, Najaf 54001, Iraq

**Keywords:** psychiatry, mood disorders, major depression, neuroimmune, oxidative stress, precision medicine models

## Abstract

Recent studies have proposed valid precision models and valid Research and Diagnostic Algorithmic Rules (RADAR) for recurrent major depressive disorder (MDD). The aim of the current study was to construct precision models and RADAR scores in patients experiencing first-episode MDD and to examine whether adverse childhood experiences (ACE) and negative life events (NLE) are associated with suicidal behaviors (SB), cognitive impairment, and phenome RADAR scores. This study recruited 90 patients with major depressive disorder (MDD) in an acute phase, of whom 71 showed a first-episode MDD (FEM), and 40 controls. We constructed RADAR scores for ACE; NLE encountered in the last year; SB; and severity of depression, anxiety, chronic fatigue, and physiosomatic symptoms using the Hamilton Depression and Anxiety Rating Scales and the FibroFatigue scale. The partial least squares analysis showed that in FEM, one latent vector (labeled the phenome of FEM) could be extracted from depressive, anxiety, fatigue, physiosomatic, melancholia, and insomnia symptoms, SB, and cognitive impairments. The latter were conceptualized as a latent vector extracted from the Verbal Fluency Test, the Mini-Mental State Examination, and ratings of memory and judgement, indicating a generalized cognitive decline (G-CoDe). We found that 60.8% of the variance in the FEM phenome was explained by the cumulative effects of NLE and ACE, in particular emotional neglect and, to a lesser extent, physical abuse. In conclusion, the RADAR scores and plots constructed here should be used in research and clinical settings, rather than the binary diagnosis of MDD based on the DSM-5 or ICD.

## 1. Introduction

Recent research has shown that there are no valid models of major depressive disorder (MDD) and that there is no replicable and cross-validated model that can be used as an outcome variable in biomarker research [1,2]. When discussing depression, it seems as though psychiatrists cannot understand one another and speak different languages. Different concepts of models (from folk psychology to molecular psychiatry) and subtypes or subclasses (MDD, melancholia, recurrent depressive disorder, dysthymia, double depression, reactive depression, vital depression, and treatment-resistant depression) rule in chaos [1,3]. As a result, research on MDD is plagued by severe noise, resulting in a cacophony of models, labels, and subtypes without a solid consensus among psychiatrists.

In addition, folk psychologists and sociologists ascertain that depression is a “boundary experience” and that “psychiatry transformed normal sorrow into depressive disorder and contributed to the medicalization of feeling blue, grieving, demoralization, and sadness” [4,5,6,7]. Consequently, in psychiatric research, severe medical phenotypes and common emotional distress responses are grouped together, resulting in an entirely heterogeneous MDD study population [1,3]. The Western gold standard to diagnose MDD using either the DSM [8] or ICD [9] criteria exacerbates this chaos [1,3]. Indeed, the DSM and ICD definitions of mood disorders lack psychiatrists’ consensus, are unreliable, and are invalid, leading to misdiagnoses and misclassifications [1,2]. In addition, the top-down dogmatism of the DSM/ICD definitions precludes inductive (because it is top-down) and deductive (because the criteria are indisputable unless the dispute is made by the same group of professionals) reworking of the criteria [1]. As a result, their use as an explanatory variable in statistical analyses is not only conceptually flawed but also results in a multitude of errors and inaccurate conclusions [1,3].

Recently, we have created a new supervised and unsupervised machine learning clinimetric approach called “precision nomothetic psychiatry”, which allows us to build new pathway phenotypes and endophenotype classes [1,2,3,10,11,12]. With the help of those methods, we are able to develop (i) bottom-up, data-driven nomothetic psychiatry models of MDD, and (ii) new pathway phenotypes of MDD in the form of phenome (the symptomatome of MDD) scores and a recurrence of illness (ROI) index, based on the recurrence of lifetime suicidal ideation (SI) and attempts (SA) and depressive episodes [1,2,3,10,11,12]. Previous studies by our team have established that adverse childhood experiences (ACE) are causal factors in ROI, the phenome, and lifetime and current suicidal behaviors (SB) [1,2,13,14]. While genetics and adverse outcome pathways play a key role in the development of MDD, ACE and negative life events (NLE) in the year before the onset of depression also contribute to MDD [13,14,15,16].

Recently, we also provided algorithms for computing Research and Diagnostic Algorithmic Rule (RADAR) scores for ACE, ROI, lifetime and current SB, phenome scores, and lifetime trajectory (which is a composite of ACE, ROI, SB, and phenome scores) of mood disorder patients [3,17]. We showed how to plot all of these different features of depression as RADAR scores in a two-dimensional RADAR or spider graph, whereby a patient’s data can be visualized, much like a fingerprint, which aids in quickly evaluating the patient’s features [3,17]. By consolidating multiple RADAR scores into one simple graph, our method demonstrates how simplistic and minimal the DSM-5 and ICD diagnoses really are by reducing all features into an unreliable, binary MDD diagnosis. Specifically, we argued that clinicians and psychiatric researchers should always use the derived RADAR scores reflecting ACE, ROI, SB, neurocognitive, phenome scores, and lifetime trajectory scores, rather than relying on invalid binary diagnoses [3,17]. ROI is the most important factor in this precision model because it determines the severity of current SB and the phenome in both the acute and partially remitted phases of depression [1,11,18]. Our model was developed using Brazilian and Thai patients in the acute and remission phases of recurrent depression and patients who showed a wide range of ROI scores. Open questions include whether NLE and ACE increase the risk of developing new-onset MDD and whether our nomothetic model and RADAR scores can be computed for first-episode MDD in other countries and cultures.

Hence, this study was carried out to ascertain (a) whether a valid nomothetic model and valid RADAR scores (excluding ROI scores) can be computed in Iraqi patients experiencing their first depressive episode; (b) whether the combined effects of ACE and NLE increase vulnerability to new-onset depression; and (c) whether there are any differences in RADAR scores between first- and second-episode MDD.

## 2. Methods

### 2.1. Participants

In the present case–control study, 98 consecutively admitted patients with a major depressive disorder (MDD) were recruited between February 2021 and March 2022 at the psychiatric unit of Al-Hakeem Hospital in the Al-Najaf region, Iraq. Due to the exclusion criteria, eight of these patients were excluded (six patients due to comorbidities, including type 1 diabetes mellitus and chronic kidney disease, and 2 patients withdrew). A senior psychiatrist diagnosed MDD based on the DSM-5 [8] criteria and selected 71 patients with MDD, more specifically first-episode MDD that was moderate or severe without psychotic symptoms, and 19 individuals with recurrent (two episodes) MDD that was moderate or severe without psychotic features. The primary focus of the current study was to build a nomothetic model and construct RADAR scores and graphs in first-episode patients. A secondary aim was to compare the RADAR scores among first-episode and second-episode patients. All patients with MDD were in the acute phase of the disease, and none exhibited complete or partial remission. The mean (SD) duration of illness for patients with first episode of MDD was 2.5 (±0.3) months. A total of 27 of the 71 first-episode patients were drug naive, whereas the other patients received medication for at least 3 weeks: 42 patients were given fluoxetine, 10 were given amitriptyline, 8 were given escitalopram, 12 were given mirtazapine, and 9 were given olanzapine. The same senior psychiatrist also recruited forty apparently healthy controls from the same catchment area among medical staff or their friends and the patients’ friends. The patients and controls were excluded if they had any other DSM-5 axis-1 disorders, such as autism spectrum disorders; dysthymia; schizophrenia; bipolar disorder; substance use disorders, except for tobacco use disorder (TUD); major anxiety disorders, including generalized anxiety disorder and panic disorder; post-traumatic stress disorder; and obsessive–compulsive disorder. In addition, the controls with a lifetime diagnosis of MDD or a family history of depression, bipolar disorder, substance use disorders, or psychosis were excluded. We also excluded pregnant and lactating women and subjects with (a) neurodegenerative or neuroinflammatory disorders, including stroke, multiple sclerosis, Parkinson’s disease, and Alzheimer’s disease; (b) chronic liver and kidney disorders; and (c) (auto)immune diseases, including psoriasis, rheumatoid arthritis, inflammatory bowel disease, cancer, type 1 diabetes, scleroderma, moderate and critical COVID-19, and rheumatoid arthritis. In addition, subjects treated with immunosuppressive or immunomodulatory drugs or therapeutic doses of antioxidants or omega-3 supplements were ineligible for participation.

### 2.2. Measurements

The senior psychiatrist conducting the study collected demographic (marital, occupational, and educational) and clinical data (duration of the index episode, age of onset, and prior COVID infection and severity of infection) using a semi-structured interview. He utilized the DSM-5 diagnostic criteria [8] to identify MDD patients and exclude those with other axis-1 diseases. We assessed ACE using the Adverse Childhood Experience (ACE) Questionnaire [19], which assesses 10 major abuse, neglect, and household dysfunction domains as present or not present, including ACEQ1: emotional abuse; ACEQ2: physical abuse; ACEQ3: sexual abuse; ACEQ4: emotional neglect; ACEQ5: physical neglect; ACEQ6: divorce; ACEQ7: violent behavior; ACEQ8: substance abuse; ACEQ9: mental illness; and ACEQ10: incarcerated relative. Negative life events (NLE) in the year prior to the onset of depression were assessed using the Negative Life Events scale [20], and we considered the following items to be relevant: serious accidents, death of a family member or close relative, divorce or separation, seeing fights, abuse or violent crime, trouble with the police, and a member of the family sent to jail. Suicidal behaviors were assessed using two items of the Columbia Suicide Severity Rating Scale: frequency of suicidal ideation and frequency of suicidal attempts (C-SSRS) [21]. Cognitive functioning was assessed using the Verbal Fluency Test (VFT) [22] to assess word fluency and semantic memory; the Mini-Mental State Examination (MMSE) [23]; and the Clinical Dementia Rating (CDR) scale [24], which assesses six domains on a 0–3 point scale, including memory, orientation, judgement and problem solving, community affairs, home and hobbies, and personal care. Nevertheless, we did not use the CDR as proposed by Morris [24]; instead, we used it to sum up memory, orientation, and judgement, coupled with the VFT and the MMSE, to examine whether we could extract one meaningful and validated principal component (PC) (see below). We also intended to examine the other 3 items (community affairs, grooming, and hobbies) as possibly contributing to the phenome scores of MDD. However, we could not find any relevance of these measurements. The severity of depression and anxiety was assessed using the Hamilton Depression (HAMD) and Anxiety (HAMA) Rating Scales [25,26]. We used the FibroFatigue scale [27] to assess the severity of fibromyalgia, chronic fatigue syndrome-like symptoms, and physiosomatic symptoms. The diagnosis of TUD was made according to the DSM-5 criteria.

### 2.3. Statistics

We utilized analysis of variance (ANOVA) or the Mann–Whitney U test to compare continuous variables and analysis of contingency tables (χ^2^ test) to compare nominal variables between groups. Multiple comparisons among group means were examined using Fisher’s protected least significant difference. Using Pearson’s product-moment correlation coefficients, we examined the associations between scale variables. Using multiple regression analysis, the effects of the explanatory variables (ACE, NLE, age, sex, and education) on the dependent variables (e.g., symptomatome and phenome scores) were examined. In addition, we used a forward stepwise automatic regression method using *p*-values of 0.05 to enter and 0.06 to remove. We generated the standardized β coefficients with t-statistics and exact *p*-values for each of the explanatory variables in the final regression model, in addition to the F-statistics (and *p*-values) and total variance (R^2^ or partial eta squared used as effect size) explained by the model. Collinearity and multicollinearity were investigated using tolerance (cut-off value 0.25), the variance inflation factor (cut-off value > 4), and the condition index and variance proportions from the collinearity diagnostics. The White and modified Breusch–Pagan tests were used to verify the presence of heteroskedasticity. All the above tests were two-tailed, and an alpha value of 0.05 was deemed statistically significant. In order to normalize the distribution of the data, some variables were first converted via transformations, including logarithmic or rank-based inversed normal (RINT) transformations.

We used principal component (PC) analysis to check whether a set of MDD features could be reduced to one meaningful PC. To be acknowledged as a validated PC, the first PC must account for >50% of the variance in the data; all loadings on this factor must be >0.7; and the factoriability of the correlation matrix must be satisfactory, as determined by the Kaiser–Meyer–Olkin (KMO) test (KMO should be >0.6). Moreover, the Bartlett’s test of sphericity (*p* should be <0.05) and the anti-image matrix should be sufficient. All statistical tests were conducted using Windows version 28 of the IMB SPSS application.

Path analysis using partial least squares (PLS) analysis (SmartPLS) [28] was used to predict the final outcome (output) variable, namely the phenome of mood disorders, using a set of independent (input) variables, including ACE, NLE, and neurocognitive impairments. Additionally, the model accounts for mediated effects (e.g., the effects of ACE on the phenome are mediated by G-CoDe). The variables were either entered as single indicators or as latent vectors (e.g., a factor extracted from all symptom domains). Complete PLS path analysis was only performed if the inner and outer models met the following predetermined quality criteria: (a) the latent vectors of the outer models demonstrate high convergent and construct validity, as indicated by Cronbach’s alpha > 0.7, composite reliability > 0.8, rho A > 0.8, and high loadings (>0.7) at *p* < 0.0001 of the indicators of the latent vectors, and (b) the overall model fit, namely the standardized root mean square (SRMR), is <0.08. PLSPredict and the cross-validated predictive ability test (CVPAT) were used to evaluate the replicability of the final PLS model. The Q^2^ values were used to estimate whether the model’s prediction error is significantly smaller than the prediction error of the naive and linear regression model benchmarks. In addition, we employed confirmatory tetrad analysis (CTA) to ensure that the latent constructs were not incorrectly specified as a reflective model. Permutation and multi-group analysis (MGA) were used to investigate whether the predefined groups (including men versus women, TUD versus non-sTUD, and drug-naive versus medicated) show significant differences in the parameter estimates [28,29,30,31] and to delineate whether the models originate from a common population. Invariance assessment of composite models (MICOM) was used to evaluate “configural and compositional invariance, and the equality of composite mean values and variances” [28,29,30,31,32]. Using the heterotrait-monotrait (HTMT) ratio with a cutoff value of >0.85, the discriminant validity of the constructs was determined. If all the previously mentioned estimates of model fit met the predetermined criteria, we conducted a complete PLS path analysis with 5000 bootstrap samples and calculated the path coefficients (with exact *p*-values) and specific indirect, total indirect (i.e., mediated), and total effects.

## 3. Results

### 3.1. Construction of Different RADAR Scores

Based on our previous publications [1,2,3,17], we computed several symptom subdomains or RADAR scores as sums or z unit-based composite scores summing up the scores of different items.Because it was impossible to find validated PCs in the ACE data, the total ACE score was calculated as the sum of the 10 ACE indicators. We also entered all ACE indicators as single indicators in the analyses (except Q3, Q5, and Q8, which showed virtually no variance).The NLE score was computed as the presence of any NLE item encountered over the last year. We examined whether ACE and NLE could best be presented separately, as an interaction term ACE × NLE, or as the sum of different adverse events (AE). In the regression analysis, it was most appropriate to enter separate ACE and NLE scores. Furthermore, the interaction pattern was also significant in the regression analysis. In the ANOVAs, the sum of ACE + NLE (yes/no) was most appropriate (labeled as adverse events or AE). Consequently, the AE score was computed as the RINT of total ACE + NLE. Using a visual binning method, the study sample was divided into three groups, namely a group with few AE (<−0.60), a group with some AE (≥−0.60 to 0.58.9), and another group with many AE (≥0.59).The pure depressive domain score was computed as a z-based composite score: the sum of the z scores of depressed mood + feelings of guilt + loss of interest (HAMD) + sadness (FF) + depressed mood (HAMA).The pure anxiety domain score was computed as a z-based composite score: the sum of anxious mood + tension + fears + anxiety behavior at interview (all HAMA items) + anxiety, psychological (a HAMD item). Both the pure depression and anxiety scores were processed as RINT scores.The pure physiosomatic symptom domain score was computed as a z unit-based composite score based on the sum of the z scores of anxiety somatic + gastrointestinal + genitourinary + hypochondriasis somatic sensory + cardiovascular + gastrointestinal (GIS) + genitourinary + autonomic symptoms + respiratory symptoms (all HAMA symptoms) + muscle pain + muscle tension + fatigue + autonomic + gastro-intestinal + headache + malaise (all FF scale items) + anxiety somatic + somatic gastro-intestinal + general somatic + genital symptoms + hypochondriasis (all HAMD items).The melancholia domain score was computed as the sum of insomnia late + psychomotor retardation + psychomotor agitation + loss of weight + diurnal variation.The insomnia domain score was calculated as a z unit-based composite score computed as the sum of the z scores of insomnia early + insomnia middle + insomnia late (all HAMD items) + sleep disorders (FF item) + insomnia (HAMA item).The subjective cognitive impairment (SCI) score was computed as a z unit-based composite calculated as concentration disorders + memory disturbance (FF scale) + intellectual problems (HAMA item) + cognition (HAMD item).The phenome1 score was computed as a PC extracted from the pure depression, pure anxiety, physiosomatic symptom, melancholia, insomnia, and SCI symptom domains. The first PC showed an adequate model fit, with AVE = 79.21%, Cronbach’s alpha = 0.921, and factor loadings that were all >0.786 (KMO = 0.885, Bartlett’s χ^2^ = 708.05, df = 15, *p* < 0.001).The suicidal behavior (SB) score was computed as a composite score calculated as frequency of suicidal ideation + frequency of suicidal attempts + suicidal ideation (HAMD item).The phenome2 score was computed as the first PC score extracted from the pure depression, pure anxiety, physiosomatic symptom, melancholia, insomnia, and SCI symptom domains and the SB score. The first PC showed an adequate model fit, with AVE = 77.06%, Cronbach’s alpha = 0.930, and factor loadings that were all >0.770 (KMO = 0.904, Bartlett’s χ^2^ = 826.60, df = 21, *p* < 0.001).In accordance with recent findings regarding schizophrenia showing that one factor reflecting a generalized cognitive decline (G-CoDe) could be extracted from several cognitive tests results [33], we examined whether one PC could be extracted from different neurocognitive tests for MDD. Indeed, we were able to extract a G-Code construct from the MMSE, the VFT, and the sum of 3 CDR item scores (memory + orientation + judgement). This first PC showed an AVE = 63.65%, Cronbach’s alpha = 0.691, and factor scores > 0.763 (KMO = 0.666, Bartlett’s χ^2^ = 62.98, df = 3, *p* < 0.001).The phenome3 score was computed as a PC score extracted from all 6 abovementioned symptom domains, SB, and G-CoDe. This first PC showed an adequate model fit, with AVE = 75.84%, Cronbach’s alpha = 0.939, and factor loadings that were all >0.776 (KMO = 0.918, Bartlett’s χ^2^ = 945.31, df = 28, *p* < 0.001).The ROI score in the total study group (thus, with second-episode patients included) was computed as the RINT transformation of a composite score built using frequency of suicidal ideation, frequency of suicidal attempts, and number of episodes (the ROI score could only be used when analyzing the total study sample).The lifetime trajectory score was assessed as the first PC score extracted from the AE, SB, G-CoDe, and phenome1 scores. This first PC showed an adequate model fit, with AVE = 70.59%, Cronbach’s alpha = 0.860, and factor loadings > 0.794 (KMO = 0.821, Bartlett’s χ^2^ = 198.21, df = 6, *p* < 0.001).

### 3.2. Features of Study Groups with Low, Some, and Many AE

Table 1 shows the socio-demographic and clinical variables in the subjects divided into those with few, some, and many AE. There were no significant differences in age, sex ratio, BMI, education, marital status, TUD, and prior COVID-19 infection between the three study groups. There was a significant association between this division and the diagnosis of first-episode major depression. The subjects with some AE and many AE had lower G-CoDe and SCI scores than the subjects with few AE. There were significant differences in SB, all symptom domains (except SCI), and phenome1, phenome2, and phenome3 scores between the three study groups. The scores increased from the group with few AE to the group with some AE group to the group with many AE. The subjects with NLE reported a significantly higher mean (SD) number of ACE (1.73 ± 0.98) when compared to those without NLE (0.98 ± 1.09) (Mann–Whitney U test: *p* < 0.001).

### 3.3. Correlations between ACE, AE, and Symptom Domains

Table 2 shows the correlation matrix between ACE and AE and the symptom domains measured in the controls and first-episode MDD patients. ACE and AE were significantly and negatively correlated with G-CoDe and positively with all other symptom domains.

### 3.4. Multiple Regression Analysis with Phenome Features as Dependent Variables

Table 3 shows the results of the multiple regression analyses with the phenome features as the dependent variables and ACE, NLE, and AE as the explanatory variables, while allowing for the effects of socio-demographic characteristics. Model #1 shows that 35.6% of the variance in the G-CoDe scores is explained by AE, and that a combination of ACEQ2, ACEQ4, ACEQ6, and NLE explains up to 45.8% of the variance in neurocognitive impairments (model #2). Figure 1 shows the partial regression of G-CoDe on AE. The latter explains (model #3) 27.4% of the variance in SB, while ACEQ4, ACEQ10, and NLE explain 32.9% of the variance (model #4). Figure 2 shows the partial regression of the SB scores on AE. We found that 35.2% of the variance in the phenome2 scores (model #5) is explained by the regression on AE and gender (higher in males), and that ACEQQ4, NLE, and ACEQ2 are the best predictors of the phenome2 scores, explaining 37.3% of the variance (model #6). Figure 3 shows the partial regression of the phenome1 scores on AE. Table 1 does not list the regression concerning phenome1 because the results are similar to those of phenome2. This is because the phenome scores are highly significantly intercorrelated (all r > 0.96, *p* < 0.0001). We found that 37.5% of the variance in the phenome3 scores (model #7) is explained by the regression on AE and male sex (all positively associated) and that male sex, ACEQ2, ACEQ4, and NLE predict the phenome3 scores and explain 42.9% of the variance (model #8). In model #9, we added the interaction term Q4 X NLE, which contributes significantly to the phenome3 scores (inversely associated).

### 3.5. RADAR Plots

The RADAR plot in Figure 4 displays the RADAR scores for two patients (MDD1 and MDD2). All scores are expressed as z scores with the mean of the healthy controls set to zero. The common center point is established as the mean values (set to zero) of all feature scores of the healthy controls. Therefore, the graphs show the relative position of the feature scores of the patients versus the mean values of the controls in standard deviations. The RADAR plot provides 14 RADAR or feature scores displayed on 14 radial axes, each corresponding to a feature. The latter are ordered along the lifetime trajectory of the patients, starting with ACE, then AE, G-CoDe, symptomatome domain features, SB, phenome scores, and finally lifetime trajectory score. The radial axes in the RADAR plot are joined in the middle of the figure (zero feature scores of the controls) and are joined by angular axes that divide the plot into grids, which show the variation in feature ratings of the two patients versus the healthy controls. This figure shows that the RADAR charts of both patients are quite different, in particular the ACE, AE, depression, anxiety, physiosomatic, SB, all phenome, and lifetime trajectory scores. Figure 5 shows the RADAR plot for two other MDD patients, MDD3 and MDD4.

### 3.6. Results of PLS Analyses

The first PLS model considered the latent vector extracted from the phenome2 features as the dependent variable, and G-CoDe, AE, and ACE as the explanatory variables. Moreover, G-CoDe was entered as a mediating variable that was allowed to mediate the effects of ACE and AE on the phenome2 scores. With an SRMR of 0.051, the model quality fit is more than adequate. The convergent reliability is more than adequate for the phenome2 scores (0.733) and the G-CoDe scores (0.609). The composite reliabilities of both constructs are more than adequate, namely 0.929 for the phenome2 scores and 0.746 for the G-CoDe scores. PLSPredict shows that all Q^2^ predicted values for the manifest and latent variables are positive, indicating that the model outperforms the most naive benchmark. Application of the CVPAT framework in PLSpredict examined the predictive reliability of the two endogenous constructs; the result shows that they have significantly lower average loss (t = 3.04, *p* = 0.003) and, thus, higher predictive validity than the indicator-average benchmark. Nevertheless, the model is not valid as no discriminatory validity is obtained: the HTMT ratio of 0.993 shows that the G-CoDe scores cannot be discriminated from the phenome2 scores. Consequently, we built a new model, including the G-CoDe scores in the phenome3 factor, as shown in Figure 6. The model fit is adequate, with an SRMR of 0.042. The convergent and composite reliabilities of the phenome3 scores are more than adequate, with an AVE = 0.701, rho_A = 0.944, and Cronbach’s alpha = 0.939. PLSpredict shows that all manifest and latent variables’ Q^2^ values are positive. The CVPAT shows that the average loss differences of PLS-SEM versus the indicator average (t = 5.07, *p* < 0.001) and the linear model (t = 2.50, *p* = 0.014) are significant, indicating a strong (t = 3.04, *p* = 0.003) predictive validity of the construct. The PLS path analysis performed using 5000 bootstraps shows that 60.8% of the variance in phenome2 is explained by the regression on ACEQ2, ACEQ4, and NLE and that the interaction (moderation) between Q4 and NLE is significant and shows an inverse effect. The PLS multigroup analysis and permutation analysis show no differences in the model parameters between men and women, or between smokers and non-smokers.

### 3.7. Effects of Drug State

Since some of the first-episode patients were drug-naive, we were able to decipher possible differences between drug-naive and medicated patients. The latter were treated with antidepressants or atypical antipsychotics for at least three weeks. Introducing the medication status (drug-naive versus use of psychotropic drugs) showed that there was no significant effect of drug state on phenome3 (t = 1.76, *p* = 0.079). The MICOM showed that the permutation *p*-values for the variables were non-significant, indicating computational invariance when comparing drug-naive and medicated patients. The mean original differences fell within the 2.5% and 97.5% limits, suggesting invariance in composite equality. Consequently, we performed PLS MGA and permutation MGA. These analyses showed no significant differences in model parameters, including pathway coefficients and outer loadings (bootstrap MGA, parametric test, and Welch–Satterthwait test), and quality criteria (explained variance, AVE, etc.) between drug-naive and medicated patients. The ANOVAs showed increased scores of pure depression (F = 4.10, df = 1/69, *p* = 0.047), SB (F = 67.96, df = 1/69, *p* < 0.001), phenome2 (F = 7.43, df = 1/69, *p* = 0.008) and phenome3 (F = 6.59, df = 1/69, *p* = 0.012) among medicated patients compared to drug-naive patients (see Figure 7). In contrast, sleep disorders were better in the medicated group (F = 5.34, df = 1/69, *p* = 0.024).

### 3.8. Features of First Episode versus Second Episode in MDD

Finally, we examined differences between the 71 first-episode MDD patients and the 19 second-episode MDD patients. Table 4 shows the differences between the two groups and the healthy controls. We found that pure depressive symptoms and lifetime trajectory scores were significantly higher in the second-episode patients than in the first-episode patients. Table 2 shows the associations between ACE/AE and different RADAR scores in this study sample. Table 3 shows that a considerable part of the variance in the ROI scores (55.5%) is explained by ACEQ4, NLE, and the interaction pattern between ACEQ4 and NLE (negative effect).

## 4. Discussion

### 4.1. RADAR Scores and RADAR Plots

The first major finding of this study is that we were able to create RADAR scores and a RADAR plot for first-episode depression, representing the phenome, SB, G-CoDe, and lifetime trajectory scores. This agrees with our prior studies, indicating that such scores may be calculated for patients with recurrent mood disorders and that the scores can be displayed in a RADAR plot [3,17]. The latter enables us to display the mean differences in all features as standard deviations between the patients and the controls, as well as a patient-specific profile or fingerprint.

Firstly, the current study found that one factor could be extracted from different symptom domains (including pure depression, anxiety, and physiosomatic symptom, melancholia, insomnia, and SCI domains), indicating that these distinct symptom profiles are the manifestations of a shared or common core, i.e., the phenome of first-episode depression. This suggests that these symptom domains should be seen as highly intercorrelated domains driven by a shared pathophysiology underlying these phenome presentations. As we have previously demonstrated [1,3,12,13], sensitization in immunological and growth factor networks, activated oxidative stress pathways, decreased antioxidant defenses, enhanced bacterial translocation, and autoimmune responses are linked with phenome scores in recurrent unipolar and bipolar depression.

Secondly, our results showing that SB are part of the phenome of first-episode depression confirm our previous research that SB can be expressed as RADAR scores and are part of the phenome of recurrent depression and mood disorders [1,3]. This again demonstrates that the same pathophysiology underlies both the phenome and SB. Combinations of immune-inflammatory and nitro-oxidative pathways are associated with suicidal ideation and suicidal attempts in people with mood disorders, according to two recent systematic reviews and meta-analyses [34,35]. In contrast to recurring MDD and mood disorders, our study on first-episode depression could not incorporate the ROI index into the analysis. In this regard, it is intriguing to notice that the second-episode patients had higher pure depressive and lifetime trajectory ratings than the first-episode patients. Thus, it is possible to conclude that the deteriorating influence of ROI on the phenome [17] is already present in the second episode.

Thirdly, we discovered that a general impairment in cognitive abilities during the acute phase of first-episode depression constitutes an additional component of the phenome and, as such, should be considered a manifestation of the phenome of depression. We were able to extract one latent vector from the VFT, MMSE, and memory, judgment, and orientation test scores. This showed that people with severe first-episode depression had a more generalized cognitive decline. In a previous study, executive functioning, semantic and episodic memory, recall, strategy utilization, rule acquisition, emotional recognition, visual sustained attention, and attentional set-shifting were used to develop a G-CoDe score for schizophrenia [33]. In contrast to these findings in stable-phase schizophrenia, the current investigation indicated that G-CoDe is a feature of the severity of the acute phase of depression. As a consequence, these impairments will probably normalize throughout the remitted and euthymic phases, given that inter-episode cognitive deterioration increases with ROI [17].

Overall, our method of using different RADAR scores to indicate the acute state of MDD contrasts with the current gold standard of employing a binary diagnosis (“major depressive episode, single episode”). It is clear that different numerical RADAR values convey far more information about the actual state of depression than a binary diagnosis, which is also unreliable. It follows that our quantitative RADAR scores should be employed as dependent factors in regression analyses or neural network studies, with neuro-immune and neuro-oxidative biomarkers as explanatory variables. Using a post hoc, erroneous, and unreliable higher-order binary construct (the DSM/ICD diagnosis) as an explanatory variable in *t*-tests or ANOVAs with biomarkers as dependent variables is grossly inaccurate and should be replaced by our nomothetic method [1,2,3,17].

### 4.2. ACE, NLE, and First-Episode Depression

The second major finding of this study is that the number of ACE and NLE significantly predicts G-CoDe, SB, and phenome scores. In earlier research, we discovered that ACE has a significant effect on G-CoDe, ROI, SB, and the acute- and residual-phase phenome of depression [13,14]. There is now compelling evidence that the cumulative effects of ACE are causally connected with the onset and severity of depression later in adulthood, as well as with cognitive impairments and suicidal tendencies [36,37,38,39,40,41,42,43,44]. In a recent study involving patients from Brazil, we discovered that emotional abuse and neglect, physical neglect, and physical and sexual abuse (combined into one latent vector) were significantly linked with adverse outcome pathways, ROI, cognitive deficits, and the phenome of depression [13,18]. In Thai patients, we discovered that a latent vector derived from mental neglect and trauma, physical trauma, and domestic violence predicted the phenome and ROI of depression [14]. In the latter study, sexual abuse; a factor from parental loss due to separation, death, or divorce; and a family history of mental illness were independently related to depression. This contradicts the current study’s conclusions that no latent vector could be identified from the 10 ACE scores included for Iraqi patients and that emotional neglect and physical abuse were the most important predictors.

In this work, we determined that NLE have a significant influence on G-CoDe, SB, and the phenome of first-episode depression, above and beyond the effects of ACE. It is known that NLE are associated with the onset of depression, including depressive symptoms in students and late-life depression, according to a large body of research [15,16,45,46,47]. Furthermore, there is evidence that stressful life events may lead to depression, rather than the other way around [48]. In addition, major life events play a greater role in the development of first-onset depression than in subsequent episodes [15].

Moreover, in the present investigation, we discovered that ACE and, in particular emotional neglect, might negatively affect the influence of NLE. This suggests that, although ACE and NLE have cumulative effects on the phenome, there is also a substantial interaction (moderation) effect, indicating that increased emotional neglect may reduce the influence of NLE. There are currently indications that the effects of ACE on the phenome, including SB, are mediated by sensitization of the cytokine and growth factor networks, which are reactivated in response to new immunological stimuli [14]. There is also evidence that psychological stressors can activate T helper-1 and M1 macrophage cytokine networks in humans [49,50,51,52,53].

Based on the present study’s findings that ACE and NLE have cumulative effects on the phenome of first-episode depression, we hypothesize that NLE may operate as a second hit to reactivate sensitized cytokine network. However, our results suggest that this effect depends on the number of ACE, so a higher number of ACE may be linked to NLE having a diminished effect. Since there is a strong association between ACE and NLE, it may also be that people with ACE select environments with more stressful events, distress, and violence, suggesting that part of the links between NLE and the phenome may be non-causal [16].

### 4.3. Limitations

If we had assessed the sensitization of the immune system using LPS + PHA-stimulated production of various cytokines and growth factors, this study would have been more interesting [11]. As we have now produced RADAR scores and plots for the acute period of first-episode depression, recurrent depression, and the remitted phase of depression in Iraqi, Thai, and Brazilian patients, future research should establish these scores in Western and Caucasian study groups. Another potentially complicating aspect is the medication state of some patients. Nevertheless, after including the impact of ACE and NLE in the PLS analysis, the drug state of the patients had no effect on the phenome, as determined by the regression analysis. Importantly, when comparing drug-naive individuals with treated first-episode patients, the latter demonstrated greater (albeit with small effect sizes) SB, pure depression, and phenome ratings, whereas the former demonstrated more insomnia. These results suggest that people with suicidal behaviors are more actively treated with psychotropic medications or, alternatively, that the treatments worsen these conditions. Another possible limitation is the putative inference with regard to the effects of acute COVID-19 infection, which may cause Long COVID with depression, anxiety, chronic fatigue, and physiosomatic symptoms [54]. Nevertheless, a Long COVID phenome is predicted by critical disease during the acute phase with high fever and low oxygen saturation, whereas this study excluded COVID-19 patients who had suffered from moderate or critical COVID disease. Future research should further examine the differences between first-episode and multiple-episode MDD using a larger sample of multiple-episode patients.

## 5. Conclusions

In this study, we demonstrated how to develop a validated precision model, RADAR scores, and RADAR plots for first-episode depression patients. We found that ACE and NLE are related to SB, cognitive impairments, and first-episode depression symptoms. We found that depression, anxiety, fatigue, physiosomatic symptoms, melancholia, insomnia, SB, and cognitive impairments are all caused by the same phenome factor. Importantly, the cumulative effects of ACE and NLE, as well as the pattern of interaction between emotional neglect and NLE, explained 60.8% of the variance in this phenome. The results suggest that ACE may attenuate the effects of NLE on the phenome. It is more appropriate to use RADAR graphs in clinical practice rather than an unreliable DSM or ICD diagnosis because the former provides more accurate information and additionally provides a personalized fingerprint of a patient’s status.

We have constructed RADAR graphs that contain not only clinical scores (as explained in the current study) but also diverse biomarker scores, which are altered in depression [55]. With the help of this new method, adequate clinical monitoring as well as biomarker monitoring of the disease may be performed. Secondly, our methodology makes it possible for researchers to compute new risk or susceptibility biomarkers, screening and detecting biomarkers, diagnostic biomarker panels, and prognostic biomarkers of ROI, suicidal behaviors, cognitive impairments, and the phenome of MDD [55]. In addition, using our novel methodology, we are able to calculate new predictive biomarker tools that can direct personalized therapies for MDD, ROI, or suicidal tendencies [55]. Axis-2 personality disorders were not measured in the current study. Some personality traits, such as neuroticism, are comorbid with depression and are affected by ACE [55]. Therefore, it is recommended to add personality trait RADAR scores to our RADAR plots.

## Figures and Tables

**Figure 1 brainsci-13-00714-f001:**
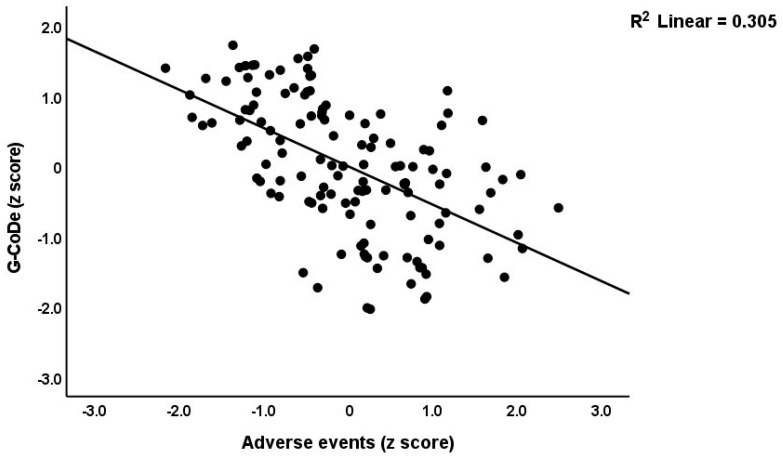
Partial regression of the generalized cognitive decline (G-CoDe) scores on adverse events during the acute phase of first-episode major depression.

**Figure 2 brainsci-13-00714-f002:**
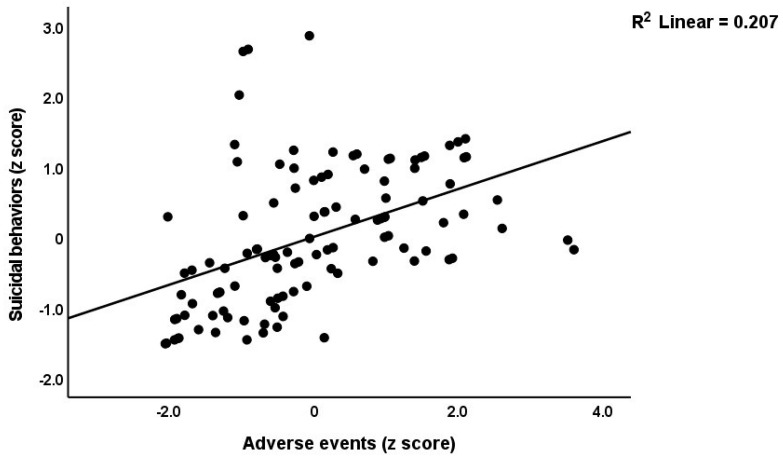
Partial regression of the suicidal behavior scores on adverse events during the acute phase of first-episode major depression.

**Figure 3 brainsci-13-00714-f003:**
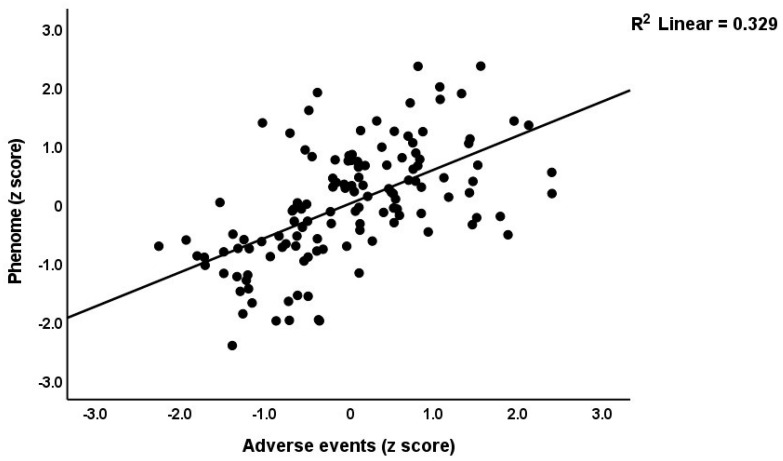
Partial regression of the phenome scores during the acute phase of first-episode depression on adverse events.

**Figure 4 brainsci-13-00714-f004:**
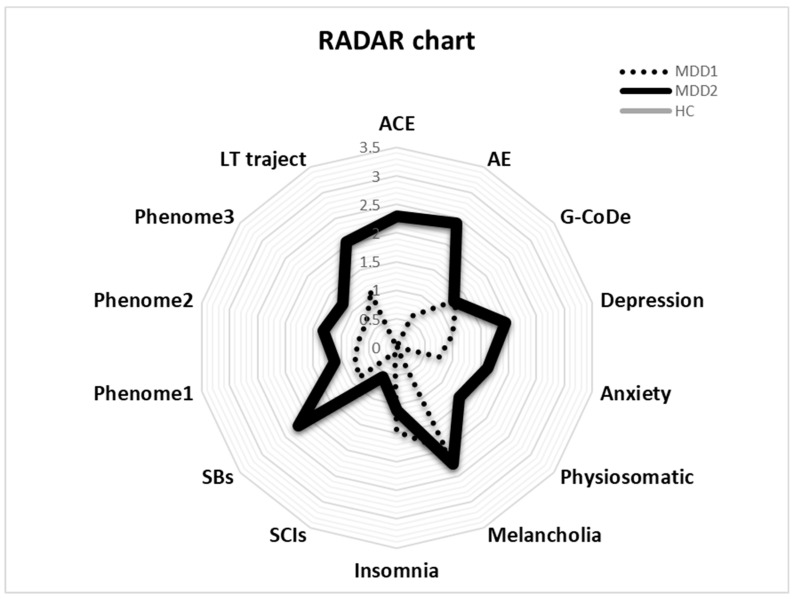
RADAR plot displaying the Research and Diagnostic Algorithmic Rule (RADAR) scores of two patients in the acute phase of first-episode depression. ACE: adverse childhood experiences; AE: adverse events; G-CoDe: general cognitive decline; SCIs: subjective cognitive impairments; SB: suicidal behaviors; phenome1: first PC extracted from all symptom domains; phenome2: same as phenome1 but includes SB; phenome3: same as phenome2 but includes G-CoDe; LT traject: lifetime trajectory.

**Figure 5 brainsci-13-00714-f005:**
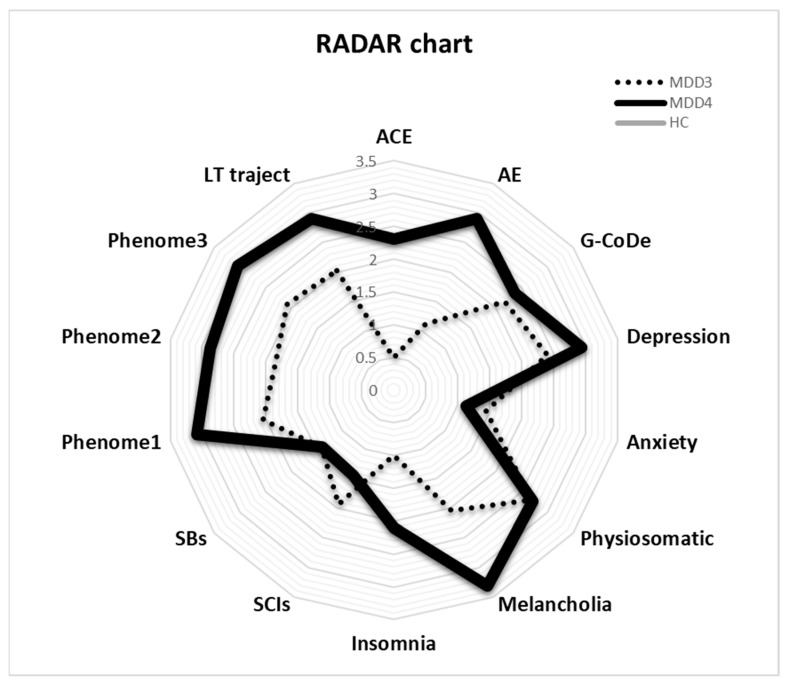
RADAR plot displaying the Research and Diagnostic Algorithmic Rule (RADAR) scores of two patients in the acute phase of first-episode major depression (MDD3 and MDD4). ACE: adverse childhood experiences; AE: adverse events; G-CoDe: general cognitive decline; SCIs: subjective cognitive impairments; SB: suicidal behaviors; phenome1: first PC extracted from all symptom domains; phenome2: same as phenomen1 but includes SB; phenome3: same as phenome2 but includes G-CoDe; LT traject: lifetime trajectory.

**Figure 6 brainsci-13-00714-f006:**
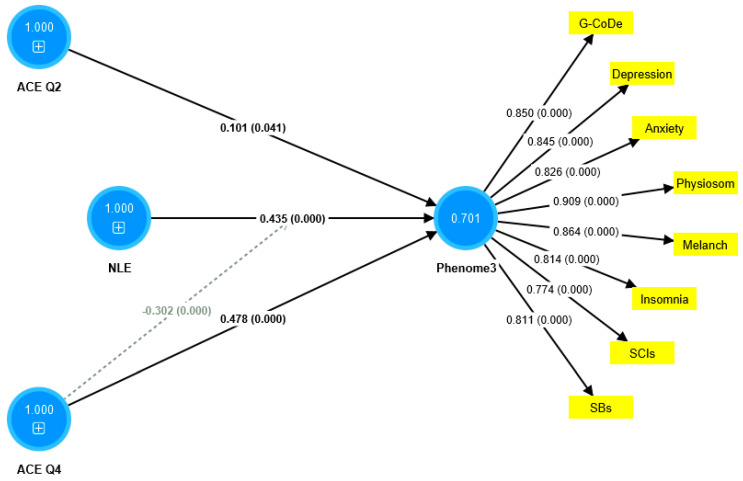
Results of partial least squares (PLS) analysis. ACE: adverse childhood experiences; NLE: negative life events; Q4: emotional neglect; and Q2: physical abuse. The indicators of phenome3 are shown as yellow rectangles, including G-CoDe: generalized cognitive decline, SCIs: subjective cognitive impairments, and SB: suicidal behaviors. The blue circles indicate single indicators and the latent vectors entered in the analysis. The pathway coefficients are shown with exact *p*-values and loadings on the phenome factor; 0.701 indicates the explained variance.

**Figure 7 brainsci-13-00714-f007:**
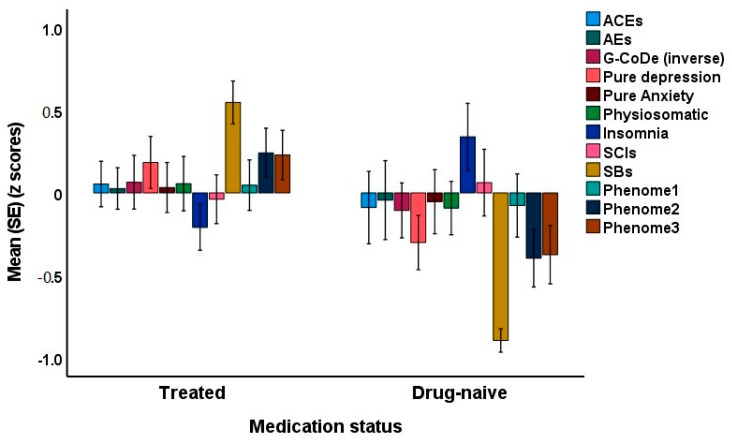
Clustered bar graph showing the mean (SE) feature scores of the acute phase of first-episode depression divided into drug-naive and treated patients. ACE: adverse childhood experiences; AE: adverse events; G-CoDe: generalized cognitive decline; SCIs: subjective cognitive impairments; SB: suicidal behaviors; LT traject: lifetime trajectory.

**Table 1 brainsci-13-00714-t001:** Socio-demographic and clinical variables in subjects divided into those with few, some, and many adverse experiences (AE).

Parameter	Few AE ^A^ n = 34	Some AE ^B^ n = 49	Many AE ^C^ n = 28	F/χ^2^	df	*p*
AE	0.26 ± 0.45 ^B,C^	1.71 ± 0.71 ^A,C^	3.71 ± 0.76 ^A,B^	2213.33	2/108	<0.001
Total ACE	0.24 ± 0.43 ^B,C^	1.22 ± 0.62 ^A,C^	2.68 ± 0.82 ^A,B^	116.22	2/108	<0.001
HC/MDD	27/7	13/36	0/28	45.45	2	<0.001
Age (years)	29.7 ± 7.4	32.4 ± 8.9	32.1 ± 8.9	1.10	2/108	0.335
Female/Male ratio	22/12	27/22	15/13	1.01	2	0.602
BMI (kg/m^2^)	25.85 ± 3.63	24.50 ± 3.79	24.44 ± 4.00	1.54	2/108	0.220
Education (years)	11/23	10/39	10/18	2.55	2	0.279
Married/Single (No/Yes)	16/18	31/18	17/11	2.30	2	0.316
TUD (No/Yes)	20/14	35/14	20/8	1.71	2	0.425
Prior COVID-19 infection (No/Yes)	19/15	33/16	15/13	1.82	2	0.402
G-CoDe	0.793 ± 0.737 ^B,C^	−0.232 ± 0.965 ^A^	−0.558 ± 0.736 ^A^	22.89	2/108	<0.001
Suicidal behaviors	−0.664 ± 0.877 ^B,C^	0.139 ± 1.024 ^A,C^	0.563 ± 0.593 ^A,B^	15.72	2/108	<0.001
Pure depression	−0.665 ± 0.871 ^B,C^	0.034 ± 0.936 ^A,C^	0.748 ± 0.673 ^A,B^	20.94	2/108	<0.001
Pure anxiety	−0.748 ± 0.667 ^B,C^	0.144 ± 0.951 ^A,C^	0.656 ± 0.858 ^A,B^	22.22	2/108	<0.001
Pure physiosomatic symtoms	−0.917 ± 0.825 ^B,C^	0.057 ± 0.970 ^A,C^	0.658 ± 0.567 ^A,B^	28.30	2/108	<0.001
Melancholia	−0.747 ± 0.789 ^B,C^	0.053 ± 0.918 ^A,C^	0.814 ± 0.649 ^A,B^	28.08	2/108	<0.001
Insomnia	−0.692 ± 0.845 ^B,C^	0.130 ± 1.024 ^A,C^	0.612 ± 0.562 ^A,B^	18.11	2/108	<0.001
SCIs	−0.685 ± 0.659 ^B,C^	0.182 ± 1.056 ^A^	0.514 ± 0.788 ^A^	15.87	2/108	<0.001
Phenome1	−0.704 ± 0.752 ^B,C^	0.032 ± 0.909 ^A,C^	0.798 ± 0.791 ^A,B^	24.93	2/108	<0.001
Phenome2	−0.681 ± 0.790 ^B,C^	0.022 ± 0.894 ^A,C^	0.789 ± 0.813 ^A,B^	23.35	2/108	<0.001
Phenome3	−0.732 ± 0.734 ^B,C^	0.080 ± 0.909 ^A,C^	0.749 ± 0.823 ^A,B^	24.40	2/108	<0.001

Data are shown as means (SD) or as ratios. F: results of analysis of variance; χ2: results of analysis of contingency tables; ^A,B,C^: pairwise comparison among group means (*p* < 0.05); BMI: body mass index; TUD: tobacco use disorder; G-CoDe: general cognitive decline; SCIs: subjective cognitive impairments; phenome1: phenome index including pure depression, anxiety, physiosomatic, melancholia, insomnia, and SCI scores; phenome2: phenome index including suicidal behaviors; phenome3: phenome index including suicidal behaviors and G-CoDe.

**Table 2 brainsci-13-00714-t002:** Correlation matrix between the number of adverse childhood experiences (ACE) alone and combined with negative life events (AE) and the symptom domains assessed in this study.

Variables	Total ACE *	AE *	Total ACE **	AE **
G-CoDe	−0.495	−0.600	−0.489	−0.586
Suicidal behaviors	0.389	0.484	0.391	0.484
Pure depression	0.515	0.559	0.502	0.548
Pure anxiety	0.455	0.565	0.434	0.528
Pure physiosomatic symptoms	0.483	0.618	0.506	0.616
Melancholia	0.510	0.597	0.534	0.617
Insomnia	0.486	0.563	0.486	0.557
SCIs	0.412	0.473	0.365	0.423
Phenome 2	0.468	0.553	0.494	0.565
Phenome 3	0.479	0.570	0.497	0.576

All significant at *p* < 0.001; * performed in controls and first-episode depressed patients (n = 111); ** performed in controls and all depressed patients combined (n = 130); G-CoDe: general cognitive decline; SCIs: subjective cognitive impairments; phenome2: phenome index including suicidal behaviors; phenome3: phenome index including suicidal behaviors and the G-CoDe.

**Table 3 brainsci-13-00714-t003:** Results of multiple regression analyses with the major symptom domains as dependent variables and adverse childhood experiences (ACE) and negative life events (NLE) as explanatory variables, while allowing for the effects of socio-demographic characteristics.

Dependent Variables	Explanatory Variables	B	T	*p*	F Model	df	*p*	R^2^
**G-CoDe**	**Model #1**				60.19	1/109	<0.001	0.356
	AE	−0.596	−7.76	<0.001				
**G-CoDe**	**Model #2**				22.36	4/106	<0.001	0.458
	ACEQ4	−0.402	−5.21	<0.001				
	NLE	−0.307	−3.94	<0.001				
	ACEQ2	−0.220	−3.01	0.003				
	ACEQ6	−0.163	−2.22	0.028				
**Suicidal behaviors**	**Model #3**				41.15	1/109	<0.001	0.274
	AE	0.524	6.42	<0.001				
**Suicidal behaviors**	**Model #4**				17.50	3/107	<0.001	0.329
	ACEQ4	0.328	3.79	<0.001				
	NLE	0.306	3.60	<0.001				
	ACEQ10	0.182	2.26	0.026				
**Phenome 2**	**Model #5**				29.31	2/108	<0.001	0.352
	AE	0.574	7.41	<0.001				
	Sex	0.165	2.13	0.013				
**Phenome 2**	**Model #6**				21.19	3/107	<0.001	0.373
	ACEQ4	0.399	4.84	<0.001				
	NLE	0.278	3.38	0.001				
	ACEQ2	0.168	2.17	0.033				
**Phenome 3**	**Model #7**				32.43	2/108	<0.001	0.375
	AE	0.589	7.74	<0.001				
	Sex	0.185	2.43	0.017				
**Phenome 3**	**Model #8**				19.89	4/106	<0.001	0.429
	ACEQ4	0.385	4.84	<0.001				
	NLE	0.287	3.64	<0.001				
	ACEQ2	0.227	3.00	0.003				
	Sex	0.173	2.30	0.023				
**Phenome 3**	**Model #9**				17.45	5/105	<0.001	0.454
	Sex	0.174	2.35	0.021				
	NLE	0.438	4.23	<0.001				
	ACEQ2	0.178	2.29	0.024				
	ACEQ4	0.563	5.01	<0.001				
	ACEQ4 X NLE	−0.311	−2.20	0.030				
**ROI (n = 130)**	**Model #10**				52.39	3/126	<0.001	0.555
	NLE	0.768	9.33	<0.001				
	ACEQ4	0.728	7.82	<0.001				
	ACEQ4 × NLE	−0.577	−5.77	<0.001				

G-CoDe: general cognitive decline; phenome2: phenome index including suicidal behaviors; phenome3: phenome index including suicidal behaviors and G-CoDe; ROI: recurrence of illness based on 2 episodes only; AE: adverse events; ACE: adverse childhood experiences; NLE: negative life events; Q4: emotional neglect; Q2: physical abuse; Q6: divorce of parents; Q10: imprisonment of a family member.

**Table 4 brainsci-13-00714-t004:** Features of healthy controls (HCs) and patients with major depressive disorder (MDD) who are divided into those with a first depressive episode (MDD #1) and those with a second depressive episode (MDD #2).

Variables	HC ^A^ n = 40	MDD #1 ^B^ n = 71	MDD #2 ^C^ n = 19	F (df = 2/122)	*p*	
Age (years)	32.1 ± 8.2	31.2 ± 8.6	32.0 ± 10.39	0.15	2/127	0.859
Female/Male ratio	23/17	41/30	12/7	2.03	2	0.904
BMI (kg/m^2^)	25.94 ± 4.18	24.30 ± 3.49	24.99 ± 2.68	2.64	2/127	0.076
Education (years)	12/28	19/52	6/13	0.24	2	0.888
Married/Single (No/Yes)	19/21	45/26	11/8	2.64	2	0.267
TUD (No/Yes)	21/19	54/17	15/4	7.65	2	0.022
Mild COVID-19 infection	23/17	44/27	10/9	0.61	2	0.736
Drug naïve (No/Yes)	40/0	44/27	19/0	28.32	2	<0.001
Total ACE	0.45 ± 10.55 ^B,C^	1.76 ± 1.06 ^A^	1.95 ± 0.85 ^A^	31.20	2/127	<0.001
AE	−1.052 ± 0.309 ^B,C^	0.428 ± 0.873 ^A^	0.613 ± 0.642 ^A^	63.42	2/127	<0.001
G-CoDe	1.170 ± 0.340 ^B,C^	−0.543 ± 0.732 ^A^	−0.435 ± 0.579 ^A^	22.89	2/108	<0.001
Suicidal behaviors	−1.025 ± 0.213 ^B,C^	0.500 ± 0.681 ^A^	0.537 ± 0.601 ^A^	101.38	2/127	<0.001
Pure depression	−1.059 ± 0.336 ^B,C^	0.411 ± 0.663 ^A,C^	0.927 ± 0.700 ^A,B^	105.56	2/127	<0.001
Pure anxiety	−1.092 ± 0.431 ^B,C^	0.550 ± 0.706 ^A^	0.408 ± 0.693 ^A^	90.41	2/127	<0.001
Pure physiosomatic symptoms	−1.270 ± 0.335 ^B,C^	0.575 ± 0.594 ^A^	0.525 ± 0.606 ^A^	165.78	2/127	<0.001
Melancholia	−1.206 ± 0.360 ^B,C^	0.487 ± 0.649 ^A^	0.719 ± 0.729 ^A^	122.17	2/127	<0.001
Insomnia	−1.209 ± 0.297 ^B,C^	0.528 ± 0.666 ^A^	0.573 ± 0.746 ^A^	120.61	2/127	<0.001
SCIs	−1.052 ± 0.258 ^B,C^	0.468 ± 0.840 ^A^	0.467 ± 0.851 ^A^	62.44	2/127	<0.001
Phenome 2	−1.109 ± 0.475 ^B,C^	0.471 ± 0.666 ^A^	0.614 ± 0.718 ^A^	92.87	2/127	<0.001
Phenome 3	−1.111 ± 0.482 ^B,C^	0.478 ± 0.668 ^A^	0.583 ± 0.715 ^A^	92.18	2/127	<0.001
Lifetime trajectory	−1.252 ± 0.341 ^B,C^	0.494 ± 0.611 ^A,B^	0.787 ± 0.601 ^A,B^	156.61	2/127	<0.001

Data are shown as means (SD) or as ratios. F: results of analysis of variance; ^A,B,C^: pairwise comparison among group means; BMI: body mass index; TUD: tobacco use disorder; ACE: adverse childhood experiences; AE: adverse events; G-CoDe: general cognitive decline; SCIs: subjective neurocognitive impairments; phenome2: phenome index including suicidal behaviors; phenome3: phenome index including suicidal behaviors and G-CoDe.

## Data Availability

The dataset (PLS models) generated during and/or analyzed during the current study will be available from Michael Maes upon reasonable request once the dataset has been fully exploited by the authors.

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
