# Peer review of "Research and Diagnostic Algorithmic Rules (RADAR) and RADAR Plots for the First Episode of Major Depressive Disorder: Effects of Childhood and Recent Adverse Experiences on Suicidal Behaviors, Neurocognition and Phenome Features"

_brainsci, 2023, doi:10.3390/brainsci13050714_

Round 1
Reviewer 1 Report
The author demonstrated how to develop a validated precision model, RADAR scores, and RADAR plots for first-episode depression patients. In a nutshell, the submitted manuscript demonstrated the sufficient enthusiasm from the study group. However, additional decryptions and rethinking of the logic and conclusion are needed.
1. There are many cloudy sentences, make the reader confusing.
2. This manuscript needs further modification and embellishments to correct language problems. There are still some grammatical and tense errors in the manuscript, which can easily lead to misunderstandings.
3. It will be beneficial to the readers if more background information (such as findings that are already established systems of MDD) is included.
4. The author should provide a brief introduction of the background in Abstract.
5. The description of the first part of results should be clearer, it is hard to understand.
6. “The phenome1 score was computed as a PC extracted from the above 6 symptom domains.” What is the 6 symptom domains? What is the correlation of phenome1 in Table 1 and Table 3?
7. What is the differences between phenome1 , phenome2 and phenome3?
8. The wrong superscript labels in “AEs” line of table 1.
9. What is the “A, B, C” in Table 1 represent? The author should describe separately.
10. In Table 1,please keep constant of the spaces between the number and superscript.
11. Rewrite the sentence “There were significant differences in SBs, all symptom domains other than SCIs, phenome2 and phenome3 scores between the three study groups with increasing scores from few ïƒ some ïƒ many AEs”.
12. “The RADAR plot in Figure 4 displays the RADAR scores for 2 patients.” Is there any mistakes? There is no RADAR plot in Figure 4.
13. “PLS path analysis performed using 5,000 bootstraps shows that 60.8% of the variance in phenome2 is explained by the regression on ACEQ2, ACEQ4 and NLE and that also the mediating effect (interaction Q4 and NLE) is significant and shows an inverse effect.” What is the mediating and inverse effect?
14. How to draw a conclusion from Figure 6?What’s the meaning of yellow or blue circle, Realize and dash line in Figure 6. This part has not been described clearly.
15. It is confusing in the labels of RADAR chart in Figures, and the “Radar or spider plot” in figure legend. The author should distinguish them.
16. Throughout the paper, the authors’ viewpoints are not entirely clear, which makes it difficult for readers to understand the innovations of this manuscript.
Author Response
Reviewer 1
The author demonstrated how to develop a validated precision model, RADAR scores, and RADAR plots for first-episode depression patients. In a nutshell, the submitted manuscript demonstrated the sufficient enthusiasm from the study group. However, additional decryptions and rethinking of the logic and conclusion are needed.
- There are many cloudy sentences, make the reader confusing.
@@ANSWER: I have reread my paper and amended some sentences.
- This manuscript needs further modification and embellishments to correct language problems. There are still some grammatical and tense errors in the manuscript, which can easily lead to misunderstandings.
@@ANSWER: I have reread the paper (also using different grammar programs and spell-checks) and corrected some typos.
- It will be beneficial to the readers if more background information (such as findings that are already established systems of MDD) is included.
@@ANSWER: This study did not aim to examine the pathways in MDD. So, I think this does not belong to the Introduction. In the discussion, we review what is known in relation to ROI, SB, the phenome, etc.
- The author should provide a brief introduction of the background in Abstract.
@@ANSWER: A brief Intro is provided, it reads:
Recent studies have proposed valid precision models and valid Research and Diagnostic Algorithmic Rules (RADAR) for recurrent major depressive disorder (MDD)
- The description of the first part of results should be clearer, it is hard to understand.
@@ANSWER: I have rephrased some parts (points 3.1, 3.1.2 3.13). See text, yellow highlights. Nevertheless, I know it is still difficult to understand for readers who do not know factor analyses, reflective and formative models, including composites.
- “The phenome1 score was computed as a PC extracted from the above 6 symptom domains.” What is the 6 symptom domains? What is the correlation of phenome1 in Table 1 and Table 3?
@@ANSWERL addressed as:
The phenome1 score was computed as a PC extracted from the pure depression and anxiety, physiosomatic, melancholia, insomnia and SCI symptom domains. Phenome1 is now shown in Table 1 (see Table 1 please). But not in Table 3 because these three phenome scores are more or less the same (all manifestations of the same underlying concept). Addressed in the text as: The regression concerning phenome1 is not shown as the results are similar to phenome2. This is because the phenome scores are highly significantly intercorrelated (all r>0.96, p<0.0001).
- What is the differences between phenome1 , phenome2 and phenome3?
@@ANSWER: was/is explained in 3.1.10, 3.1.12 and 3.1.14.
- The wrong superscript labels in “AEs” line of table 1.
@@ANSWER: corrected.
- What is the “A, B, C” in Table 1 represent? The author should describe separately.
@@ANSWER: As is/was explained: A, B, C: Pairwise comparison among group means. Now mentioned in the Statistics section: Multiple comparisons among group means were examined using Fisher’s protected least significant difference.
- In Table 1,please keep constant of the spaces between the number and superscript.
@@ANSWER: done
- Rewrite the sentence “There were significant differences in SBs, all symptom domains other than SCIs, phenome2 and phenome3 scores between the three study groups with increasing scores from few à some à many AEs”.
@@ANSWER: We rewrote this sentence, see text please.
- “The RADAR plot in Figure 4 displays the RADAR scores for 2 patients.” Is there any mistakes? There is no RADAR plot in Figure 4.
@@ANSWER: The numbering of the figures is now corrected.
- “PLS path analysis performed using 5,000 bootstraps shows that 60.8% of the variance in phenome2 is explained by the regression on ACEQ2, ACEQ4 and NLE and that also the mediating effect (interaction Q4 and NLE) is significant and shows an inverse effect.” What is the mediating and inverse effect.
@@ANSWER: Yes, this was a typo: should be moderating effect (interaction !!!). So, it is a negative interaction effect. What this means is/was explained in the text.
- How to draw a conclusion from Figure 6?What’s the meaning of yellow or blue circle, Realize and dash line in Figure 6. This part has not been described clearly.
@@ANSWER; Now described as:
The indicators of phenome3 are shown as yellow rectangles, including G-CoDe: generalized cognitive decline; SCIs: subjective cognitive impairments; SBs: suicidal behaviors. Blue circles indicate single indicators and the latent vector entered in the analysis. Shown are pathway coefficients with exact p-values and loadings on the phenome factor; 0.701 indicates the explained variance.
- It is confusing in the labels of RADAR chart in Figures, and the “Radar or spider plot” in figure legend. The author should distinguish them.
@@ANSWER: This type of chart is called a RADAR chart and we also labelled our computed scores RADAR scores. I have deleted the synonym “spider” from the legends.
- Throughout the paper, the authors’ viewpoints are not entirely clear, which makes it difficult for readers to understand the innovations of this manuscript.
@@ANSWER: I changed some sentences, and all should be clear by now.
Prof. Dr. Michael Maes, M.D., Ph.D.
Reviewer 2 Report
The authors present a very interesting model of RADAR scores and propose to use them in clinical practice as an alternative to using the DSM or ICD, as RADAR scores provide more accurate information about the patient's condition. They analyze the effects of the interactions of adverse childhood events and negative life events and how they influence the establishment of early-onset depression, suicidal behavior, or cognitive impairment.
I consider the paper to be of great interest for clinical psychiatry; it contains relevant information and is very well structured, so I just want to comment on some minor changes:
1. line 241-244: please revise the intervals you mention to form the three groups, as it seems to me that the third group called "many AEs" is contained within the second group called "some AEs", as 0.59 is within 0.60 and 0.58.
2. lines 303-310: the font size is different.
3. The authors mention that: "major life events play a more important role in the development of first-onset depression than in later episodes".
It would be important to discuss this question further from the point of view of resilience and how resilience may influence whether major life events are more important in first-onset depression than in later episodes.
Author Response
Reviewer 2
The authors present a very interesting model of RADAR scores and propose to use them in clinical practice as an alternative to using the DSM or ICD, as RADAR scores provide more accurate information about the patient's condition. They analyze the effects of the interactions of adverse childhood events and negative life events and how they influence the establishment of early-onset depression, suicidal behavior, or cognitive impairment.
I consider the paper to be of great interest for clinical psychiatry; it contains relevant information and is very well structured, so I just want to comment on some minor changes:
- line 241-244: please revise the intervals you mention to form the three groups, as it seems to me that the third group called "many AEs" is contained within the second group called "some AEs", as 0.59 is within 0.60 and 0.58.
@@ANSWER: amended as: AEs (< -0.60), one with some AEs (≥ -0.60 to 0.58), and
- lines 303-310: the font size is different.
@@ANSWER: now the font size is 10 all over the text
- The authors mention that: "major life events play a more important role in the development of first-onset depression than in later episodes". It would be important to discuss this question further from the point of view of resilience and how resilience may influence whether major life events are more important in first-onset depression than in later episodes.
@@ANSWER: I do not think this has anything to do with resilience but with our findings that with each episode (ROI) the sensitization of different pathways (immune and oxidative stress) increases, thus rendering recurrent episodes disconnected from environmental factors (see Maes et al. 2018-2023).
Prof. Dr. Michael Maes, M.D., Ph.D.
Reviewer 3 Report
In this study he Authors aimed to investigate whether (a) a valid nomothetic model 100 and valid RADAR scores (excluding ROI scores) can be computed in Iraqi patients expe- 101 riencing their first depressive episode; b) combined effects of ACE and NLEs increase vul- 102 nerability to new-onset depression; and c) whether there are any differences in RADAR 103 scores between first- and second-episode MDD.
Overall, I found the study timely, original, well conducted and scientifically sound. However, I have some minor comments aimed at improving the high quality of the paper, and these are outlined below:
- In the introduction, a brief note on the fact that MDD might be multidimensional, including several variants with different neurobiological underpinnings, should be added with appropriate references (please see and refer to following dois: 10.1080/09540261.2020.1765517 and 10.2174/1381612825666190312102444).
- How many subjects were screened, but not included for any reason (why?) or refused to participate? Please, add some more informations on this point.
- How the presence of an intellectual disability was evaluated?
- Table 1 can be omitted and condensed as text.
- Translating into “real world” clinical practice and medicine, what possible clinical shreds of evidence might arise from the present study and what the Researchers do suggest to improve clinical practice? Please add a brief paragraph on possible suggestions in terms of integrative care.
Author Response
In this study he Authors aimed to investigate whether (a) a valid nomothetic model 100 and valid RADAR scores (excluding ROI scores) can be computed in Iraqi patients expe- 101 riencing their first depressive episode; b) combined effects of ACE and NLEs increase vul- 102 nerability to new-onset depression; and c) whether there are any differences in RADAR 103 scores between first- and second-episode MDD.
Overall, I found the study timely, original, well conducted and scientifically sound. However, I have some minor comments aimed at improving the high quality of the paper, and these are outlined below:
- In the introduction, a brief note on the fact that MDD might be multidimensional, including several variants with different neurobiological underpinnings, should be added with appropriate references (please see and refer to following dois: 10.1080/09540261.2020.1765517 and 10.2174/1381612825666190312102444).
@@ANSWER: About the first paper (Full article: From dysthymia to treatment-resistant depression: evolution of a psychopathological construct (tandfonline.com)): dysthymia, neuroticism and MDD are manifestions of the same underlying concept: the phenome of depression. See Maes et al., 2023. Thus, not addressed. The authors of that paper should check this using factor analysis.
About the second paper: I do not see any reason to Introduce a paper on glutaminergic transmission and ketamine in our Introduction. The Introduction serves to introduce what is examined in the paper and is not intended to read as a review.
- How many subjects were screened, but not included for any reason (why?) or refused to participate? Please, add some more informations on this point.
@@ANSWER: Addressed in the text as:
In the present case-control study, 98 consecutively admitted patients with a major depressive disorder (MDD) were recruited between February 2021 and March 2022 at the psychiatry unit of Al-Hakeem Hospital in Al-Najaf region, Iraq. Due to exclusion criteria eight of these were excluded (six patients due to comorbidities: type 1 diabetes mellitus and chronic kidney disease; and 2 patients withdrew).
- How the presence of an intellectual disability was evaluated?
@@ANSWER: Cognitive impairments were assessed as described in the methods and 3.1.13 (a PC extracted from different test scores)
- Table 1 can be omitted and condensed as text.
@@ANSWER: It is impossible to condense such a table into a readable text. It would be extremely boring to read such a long laundry list.
- Translating into “real world” clinical practice and medicine, what possible clinical shreds of evidence might arise from the present study and what the Researchers do suggest to improve clinical practice? Please add a brief paragraph on possible suggestions in terms of integrative care.
@@ANSWER: This study showed a method to improve the clinical diagnosis of depression. The importance is now discussed and reads:
We now construct RADAR graphs which contain not only clinical scores (as explained in the current study) but also diverse biomarker scores, which are altered in depression [55]. With the help of this new method, adequate clinical monitoring as well as biomarker monitoring of the disease may be performed. Second, our methodology makes it possible for researchers to compute new risk or susceptibility biomarkers, screening and detecting biomarkers, diagnostic biomarker panels, and prognostic biomarkers of ROI, suicidal behaviors; and the phenome of MDD [55]. In addition, using our novel methodologies, we are able to calculate new predictive biomarker tools that will direct personalized therapies for MDD, ROI, or suicidal tendencies [55].
Prof. Dr. Michael Maes, M.D., Ph.D.
Reviewer 4 Report
The review of the manuscript entitled: “Research and Diagnostic Algorithmic Rules (RADAR) and RADAR plots for the first episode of major depressive disorder: effects of childhood and recent adverse experiences on suicidal behaviors, neurocognition and phenome features”
Comments for Authors:
Thank you for the valuable research you have done. The study concerns interesting topic and acceptable writing. However, there are some issues:
1) ‘Methods’ section, ‘Participants’ subsection: The authors mentioned that participants who were bipolar or showed psychotic features were excluded from the study (lines 112, 113, 125). In the following sentences, authors mentioned that among participants “9 were given olanzapine” as treatment for depression (line 121). However, Olanzapine is not a standard single medication for depression, except for bipolar depression, and generally is being prescribed in combination with anti-depressant medications for unipolar depression when shows psychotic features. Moreover, in line 108 the authors have mentioned that “90 patients with a major depressive episode (MDD) were recruited”. However, a major depressive episode may occur in bipolar disorders and it is not equal with ‘major depressive disorder (MDD)’ which is 'unipolar depression'.
2) ‘Methods’ section, ‘Participants’ subsection, lines 118, 119: The authors mentioned that “Twenty-seven of the 71 first-episode patients were drug naïve”. Were these 27 participants receiving any kind of treatments for depression?
3) ‘Methods’ section, ‘Participants’ subsection: The authors mentioned that 71 first-episode MDD patients (line 111), 19 individuals with recurrent MDD (line 112) and 40 healthy controls (line 122) were recruited. Why authors did not recruited equal participants for each group?
4) ‘Methods’ section, ‘Measurements’ subsection: It is recommended for authors to include complete description of used scales and questionnaires including the language and statistical specifications of them.
5) ‘Methods’ section, ‘Measurements’ subsection, line 158: A kind of typing and writing error has occurred and ‘Mini Mental State Examination (MMSE)’ is correct.
6) ‘Methods’ section, ‘Measurements’ subsection, line 159: It seems that a kind of typing and writing error has occurred and the abbreviation does not match with its full phrase for “Clinical Dementia Rating (CRC)”.
7) ‘Methods’ section, ‘Measurements’ subsection: Did authors perform any tool or questionnaire for assessing and excluding other psychiatric axis’s disorders including personality disorders or mental disability disorder?
8) ‘Results’ section, line 301: It is recommended to mention the complete phrases of abbreviations (e.g. TUD) while using them for the first time in the text.
Good luck
Author Response
Reviewer 4
The review of the manuscript entitled: “Research and Diagnostic Algorithmic Rules (RADAR) and RADAR plots for the first episode of major depressive disorder: effects of childhood and recent adverse experiences on suicidal behaviors, neurocognition and phenome features”
Comments for Authors:
Thank you for the valuable research you have done. The study concerns interesting topic and acceptable writing. However, there are some issues:
1) ‘Methods’ section, ‘Participants’ subsection: The authors mentioned that participants who were bipolar or showed psychotic features were excluded from the study (lines 112, 113, 125). In the following sentences, authors mentioned that among participants “9 were given olanzapine” as treatment for depression (line 121). However, Olanzapine is not a standard single medication for depression, except for bipolar depression, and generally is being prescribed in combination with anti-depressant medications for unipolar depression when shows psychotic features. Moreover, in line 108 the authors have mentioned that “90 patients with a major depressive episode (MDD) were recruited”. However, a major depressive episode may occur in bipolar disorders and it is not equal with ‘major depressive disorder (MDD)’ which is 'unipolar depression'.
@@ANSWER: We only included patients with MDD (major depressive disorder) as defined in the participants section. Yes, in Iraq they appear to treat MDD with atypical antipsychotics (BTW: not only in Iraq).
2) ‘Methods’ section, ‘Participants’ subsection, lines 118, 119: The authors mentioned that “Twenty-seven of the 71 first-episode patients were drug naïve”. Were these 27 participants receiving any kind of treatments for depression?
@@ANSWER. No, they were drug-naive. Treatment started directly after blood sampling.
3) ‘Methods’ section, ‘Participants’ subsection: The authors mentioned that 71 first-episode MDD patients (line 111), 19 individuals with recurrent MDD (line 112) and 40 healthy controls (line 122) were recruited. Why authors did not recruited equal participants for each group?
@@ANSWER: The focus of this study was to construct models for first-episode MDD. The second-episode group was only included to examine if the ROI score from the first to the second episode has any meaning. So, only eight lines are devoted to this comparison. Addressed in the limitations as:
Future research should further examine the differences between first-episode and multiple episode MDD using a larger sample size of multiple episodes patients.
4) ‘Methods’ section, ‘Measurements’ subsection: It is recommended for authors to include complete description of used scales and questionnaires including the language and statistical specifications of them.
@@ANSWER: The scales we used are all commonly used scales in neuropsychiatry that do not need any introduction (HAMA/HAMD/FF/VFT?ACE etc). Moreover, all scales are/were well cited. So, no need to explain these scales or their psychometric properties. In addition, we repurposed these scales to compute our own indices, thus, no need to show the psychometric properties at all.
5) ‘Methods’ section, ‘Measurements’ subsection, line 158: A kind of typing and writing error has occurred and ‘Mini Mental State Examination (MMSE)’ is correct.
@@ANSWER: corrected.
6) ‘Methods’ section, ‘Measurements’ subsection, line 159: It seems that a kind of typing and writing error has occurred and the abbreviation does not match with its full phrase for “Clinical Dementia Rating (CRC)”.
@@ANSWER: corrected
7) ‘Methods’ section, ‘Measurements’ subsection: Did authors perform any tool or questionnaire for assessing and excluding other psychiatric axis’s disorders including personality disorders or mental disability disorder?
@@ANSWER: as explained, we excluded patients with other axis I disorders other than MDD. In the current study, a psychiatrist performed the task, which is adequate. However, we use the MINI when research assistants (not psychiatrists) do the interviews. No, we did not exclude personality disorders.
8) ‘Results’ section, line 301: It is recommended to mention the complete phrases of abbreviations (e.g. TUD) while using them for the first time in the text.
@@ANSWER: we have spelled out TUD when first mentioned in the text.
Prof. Dr. Michael Maes, M.D., Ph.D.
Round 2
Reviewer 1 Report
1. again,please keep the “RADAR” or “Radar” constant through whole manuscript.
2.In Figure 4 and Figure 5, please check the labels "RADAR chart" ,or Radar plot?
3. again, the "RADAR,Research and Diagnostic Algorithmic Rules " and The "Radar plot" might be confused, the spider plot is more clear.
4. please provide the software for the network constrcution, Cytoscape or?
Author Response
again,please keep the “RADAR” or “Radar” constant through whole manuscript.
ANSWER: now RADAR all over the text
2.In Figure 4 and Figure 5, please check the labels "RADAR chart" ,or Radar plot?
ANSWER: RADAR
3. again, the "RADAR,Research and Diagnostic Algorithmic Rules " and The "Radar plot" might be confused, the spider plot is more clear.
ANSWER: I like most RADAR GRAPHS and RADAR scores
4. please provide the software for the network constrcution, Cytoscape or?
ANSWER: added SmartPLS
Reviewer 3 Report
The paper is very interesting and worthy of publication
Author Response
The paper is very interesting and worthy of publication
@ANSWER: thank you
Reviewer 4 Report
The review of the revised version of the manuscript entitled: “Research and Diagnostic Algorithmic Rules (RADAR) and RADAR plots for the first episode of major depressive disorder: effects of childhood and recent adverse experiences on suicidal behaviors, neurocognition and phenome features”
Comments for Authors:
Thank you for the precise revision you have done. The paper is now much improved. However, there are some more comments based on your answers:
@@ANSWER. No, they were drug-naive. Treatment started directly after blood sampling.
Comment: It seems that a kind of typing and writing error has occurred in authors answer. Nothing about “blood sampling” was mentioned in the manuscript. However, it is recommended to add the proper sentence, which is about starting treatment after sampling, in the “Ethics approval and consent to participate” section of the paper.
@@ANSWER: as explained, we excluded patients with other axis I disorders other than MDD. In the current study, a psychiatrist performed the task, which is adequate. However, we use the MINI when research assistants (not psychiatrists) do the interviews. No, we did not exclude personality disorders.
Comment: Thanks for your answer. However, personality disorders can be interconnected with adverse childhood experiences (ACE) and/or suicidal behaviors (SB). Furthermore, many personality disorders are co-morbid with MDD. As, personality disorders have not been excluded; therefore, a notable bias has occurred. It is recommended to mention this point in limitations paragraph of discussion section of the manuscript.
Good luck
Author Response
Comment: It seems that a kind of typing and writing error has occurred in authors answer. Nothing about “blood sampling” was mentioned in the manuscript. However, it is recommended to add the proper sentence, which is about starting treatment after sampling, in the “Ethics approval and consent to participate” section of the paper.
@ANSWER: I do not understand. Was/is addressed in the text, see lines 115-118.
Comment: Thanks for your answer. However, personality disorders can be interconnected with adverse childhood experiences (ACE) and/or suicidal behaviors (SB). Furthermore, many personality disorders are co-morbid with MDD. As, personality disorders have not been excluded; therefore, a notable bias has occurred. It is recommended to mention this point in limitations paragraph of discussion section of the manuscript.
@ANSWER: Now addressed in the limitations as:
Axis-2 personality disorders were not measured in the current study. Some personality traits such as neuroticism are comorbid with depression and are affected by ACE [55]. Therefore, it is recommended to add personality trait RADAR scores to our RADAR plots.
Round 3
Reviewer 1 Report
none
Author Response
Comments and Suggestions for Authors: none @ANSWER: than you